# A Qualitative Investigation on COVID-19 Vaccine Hesitancy in Neurodivergent Communities

**DOI:** 10.3390/vaccines11050895

**Published:** 2023-04-25

**Authors:** Laila N. Khorasani, Asal Bastani, Tammy Shen, Gurlovellen Kaur, Nilpa D. Shah, Lucia Juarez, Michelle Heyman, Julie Grassian, An-Chuen Cho, Emily Hotez

**Affiliations:** 1David Geffen School of Medicine, University of California, Los Angeles, CA 90095, USA; asalbastani27@ucla.edu (A.B.); ehotez@mednet.ucla.edu (E.H.); 2Graduate School of Education, University of California, Riverside, CA 92521, USA

**Keywords:** COVID-19, vaccine hesitancy, neurodivergent, intellectual and developmental disability, thematic coding

## Abstract

Vaccine hesitancy is a major barrier to vaccination, hindering the success of vaccine efforts and thereby increasing public health risk to viral diseases, including COVID-19. Neurodivergent (ND) individuals, including individuals with an intellectual and/or developmental disability, have demonstrated a heightened risk of hospitalization and death due to COVID-19, highlighting the need for further research specifically on ND communities. We conducted a qualitative analysis using in-depth interviews with medical professionals, non-medical health professionals and communicators, and ND individuals or their caregivers. Using a thematic coding analysis methodology, trained coders identified major themes according to 24 distinct codes spanning across the categories of (1) barriers to vaccination; (2) facilitators to vaccination; and (3) suggestions for improving vaccine confidence. Qualitative findings identify misinformation, perception of vaccine risk, sensory sensitivities, and structural hardship as the most significant barriers to COVID-19 vaccination. We highlight the importance of accommodations to vaccination for the ND community alongside coordinated efforts by healthcare leaders to direct their communities to accurate sources of medical information. This work will inform the direction of future research on vaccine hesitancy, and the development of programs specific to the ND community’s access to vaccines.

## 1. Introduction

As the COVID-19 pandemic abruptly disrupted daily life, neurodivergent (ND) populations became disproportionately affected [1]. ND populations include individuals who identify with having an intellectual and/or developmental disability [2]. In addition, ND individuals are at an increased risk of hospitalization and death due to COVID-19 [1,3,4]. These populations are especially affected by comorbidities, such as diabetes and hypertension, that can increase the risk of COVID-19 mortality [1,4]. 

Additionally, COVID-19 pandemic restrictions disproportionately impacted stress and behavioral difficulties within the ND community [5]. The subsequent societal changes following the onset of the pandemic made it more difficult for ND individuals to access healthcare, social, and educational resources [6]. Schools and many healthcare appointments were moved to a virtual setting, which placed an unexpected burden on parents and caregivers to care for children in ways they previously did not [7]. The resulting routine disruptions and rampant uncertainty regarding access to safe foods and medications has increased anxieties in ND populations with limited access to support [8]. 

As a result of the direct and indirect impacts of the COVID-19 pandemic on the ND community, research that can increase vaccine uptake and confidence, especially in vulnerable communities, is critical. Vaccination can directly protect ND individuals from fatal symptoms of COVID-19 should they be infected. Indirectly, vaccination can reinstate a sense of normalcy to alleviate social anxieties or stressors that have been brought on with the pandemic. Since the rollout of the first COVID-19 vaccines, hesitancy towards the novel vaccines has posed a major roadblock to the development of herd immunity conditions [9]. In a study conducted amongst autistic adults, concerns about vaccine safety were more prevalent than concerns about contracting COVID-19 [10,11]. Perceived risks and vaccine safety concerns particularly add to vaccine hesitancy. Even within the ND community, general hesitancy towards vaccines due to false beliefs that vaccines cause autism spectrum disorders may also play a role in COVID-19 vaccine specific hesitancy [12,13]. Some researchers suggest that ND individuals do not want to receive the COVID-19 vaccine not only because of its own risks, but also due to this overall vaccine hesitancy [3,10,12]. 

Misinformation correlating the COVID-19 vaccination with autism has particularly been rampant on social media, posing challenges to public health vaccine responses. ND individuals, along with other vulnerable populations, who have experienced mistreatment within the healthcare system are likely to accept online misinformation to avoid institutions that have historically marginalized their communities [10,11]. Vaccine hesitancy stemming from concerns about side effects, efficacy, and lack of trust in the government has particularly affected African American and Latino groups, as well as the parents of ND children [3,14]. These vulnerable populations could be susceptible to receiving misinformation online, highlighting a need to target such communities in public health programs [10]. 

Globally, there have been many cases in which individuals in the ND community were not prioritized for the COVID-19 vaccine, even though they have greater health risks. For instance, in the US, the Advisory Committee on Immunization Practices COVID-19 Vaccine Working Group released guidelines for which populations should be prioritized in vaccination. In these recommendations, individuals with intellectual and/or developmental disabilities were not identified as a high priority group [15]. Other countries had similar policies that did not prioritize ND individuals, which may point to the global lack of knowledge on increased risks [13,16,17]. Despite global practices, research on increased comorbidities and susceptibility factors in the ND community indicates a need for ND individuals to be a priority for COVID-19 and future vaccines [18,19].

The current literature emphasizes the critical importance of promoting vaccine access, uptake, and confidence in ND communities, especially through use of healthcare professionals as they are often the main source that families receive information about vaccines from [20,21]. However, there is a need for research that focuses on communication strategies supported by healthcare professionals with specific focus on promoting vaccine confidence in ND communities. As such, we sought to better understand the underlying factors surrounding vaccine hesitancy in the ND community, and how these hesitancies can be addressed through targeted communication efforts. Using a qualitative approach, interviewing experts and community members, we gained a more in-depth understanding of the specific challenges of the ND community that are often unavailable using quantitative methods [22]. Coupled with a growing body of literature on COVID-19 vaccine hesitancy in the general population, we aim to identify strategies to increase confidence in COVID-19 vaccines, with a specific focus on these impacts within ND communities.

## 2. Materials and Methods

### 2.1. Study Setting and Design

This investigation consists of a qualitative analysis of in-depth interviews conducted with stakeholders in the ND community. Following the interviews, each interviewee was asked a set of demographic and opinion questions related to vaccine hesitancy, their relationship with the ND community, and the role of social media in health information dissemination.

Interviews were conducted as part of Neurodiversity Health Chats (NDHC), a trainee-based initiative with the goal of COVID-19 vaccine information dissemination. Trainees across levels (baccalaureate through post-doctoral) conducted interviews as part of a team of stakeholder engagers or disseminated information directly onto several social media channels.

### 2.2. Recruitment and Participants

Most interviewees were recruited via our team’s interpersonal contacts within the UC-LEND (Leadership Education in Neurodiversity Program) clinic. Interviewees were required to identify with the ND community or have consistent contact with the ND community as a professional or caregiver. Interviewees were asked to focus on the experiences or trends that they have noticed regarding the vaccine hesitancy of ND individuals. Interviewees were recruited throughout the United States.

Many interviewees were physicians because our main goal was to understand existing vaccine confidence programs in addition to understanding the shortcomings of these programs. Given that our physician interviewee pool has extensive experience with ND communities within a healthcare setting, they are a valuable source of information to understand how vaccine programs benefit or bring disadvantages to ND communities. For instance, we reference the Needle Anxiety Program at UC-LEND extensively in Appendix A due to an interviewee’s extensive role in sharing needle anxiety resources with ND patients.

### 2.3. Population Sample

We conducted interviews with 17 individuals spanning from medical professionals, and communicators, individuals with lived experience in the ND or disability community, and caregivers (Table 1). Participants were selected via convenience sampling. One interview was conducted as a group interview due to scheduling constraints. Thus, we conducted a total of 16 interviews with a total of 17 interviewees. Two interviewees were also simultaneously undergoing formal physician or physician-scientist training. Each set of questions inquired about the attitudes that the interviewee holds or has observed toward vaccination, the barriers, and facilitators to vaccination that they have noticed, experiences with the healthcare system in relation to their ND identity, and what they hope to see in future public health efforts to raise vaccination rates. In this manuscript, gender neutral pronouns are used to refer to all participants to preserve anonymity in the analysis.

### 2.4. Data Collection

Interviews took place online via face-to-face video chat in home or workplace settings. Interviews occurred over the span of one year in two phases, with phase 1 conducted between July and September 2021, and phase 2 conducted between July and September 2022 to allow for the inclusion of multiple diverse perspectives. Only researchers in the stakeholder engagers team and participants were present at each interview. Each interviewee was only interviewed once throughout the length of this study. Interviewees consented to being visual and audio recorded. When consent was given, videos were posted online for public consumption, with minor editing for clarity. Interviews have been approved by the UCLA Institutional Review Board. Interview guides were developed in an iterative process by the stakeholder engagers team based on the expertise of individual interviewees. Representative guides are provided in the appendix, highlighting specific questions posed to the interviewees (Appendix A and Appendix B).

### 2.5. Qualitative Analysis

A set of thematic codes were developed to identify facilitators, barriers, and suggestions for vaccination confidence as a form of content analysis (Table 2, Table 3 and Table 4). Codes were derived after the stakeholder engager team analyzed a preliminary set of interviews, and they were refined after conducting a preliminary literature review. After every few interviews, codes were revised to capture predominant themes in interviews. This process was repeated until there was a consensus regarding the coding scheme [23]. These codes were binary such that “1” indicated that the theme was present, and “0” indicated that the theme was not present. Interviews were coded by two trained coders, who determined whether each theme was present or not. Participants did not offer feedback on any of the results presented in this study.

We analyzed the codes as described previously, with the exception that codes were mutually exclusive [21]. Coders trained together on a subset of interviews, revising codes and operational definitions until a consensus was achieved. The results presented include codes from one coder who coded all the interviews. Percent agreement was determined by a comparison of the codes of half of the interviews between the two coders. A consensus was not required between coders in the final coding scheme. Inter-rater agreement was calculated as described previously to confirm the reliability of the qualitative outcomes based on 50% of interviews that were dually coded [23]. The interviews included in the percent agreement calculation were randomly selected. Percent agreement was 94.27%.

## 3. Results

As part of our qualitative analysis of in-depth interviews with stakeholders throughout the ND community, we discussed both barriers and facilitators to COVID-19 vaccination alongside suggestions to improve vaccine confidence and increase vaccination rates across communities. Qualitative findings, alongside codes, operational definitions, frequencies, and selected example quotations are summarized in Table 2, Table 3 and Table 4. 

The following section summarizes popular themes amongst interviewees, with example quotes for each theme discussed.

### 3.1. Misinformation or Misconceptions about COVID-19 Vaccines

Growing misinformation about COVID-19 vaccines has been widely reported since their development [10,11]. As such, misinformation or misconceptions were among the most reported barriers to vaccination in our interview population (Table 2).
“*Research side in terms of the vaccine being a newer technology that a lot of people don’t understand again because of…the abysmal scientific literacy of our society as a whole and, it becomes really easy to latch on to things*,” Non-medical health professional.

Some participants additionally discussed the role of the media in presenting accurate information, alluding to how missing context or incorrect information could drastically affect the willingness of individuals to adhere to vaccine protocols.
“*Sometimes, that context was missing. It was just presented as the Holy Grail, or the truth*,” Medical professional.
“*You know, there’s a lot of information on where…news is often not that accurate*,” Medical professional.

A community member not directly involved in healthcare additionally highlighted the role that media coverage plays on the public perception of vaccines. This participant suggested that not just the information itself, but the manner of the information presented greatly impacted their initial willingness towards COVID-19 vaccination.
“*They didn’t really help me, like the news and stuff*,” Caregiver.

Participants often discussed misconceptions related to perceived vaccine risk. The relationship particularly between autism and vaccination has been poor since the late 1990s [24]. These perceptions appear to be related to current COVID-19 vaccine hesitancy and play a significant role in the context of the ND community’s relationship with COVID-19 vaccines [25,26].
“*Due to some sort of misconception about the risks associated with vaccines*,” ND individuals.

### 3.2. Perception of Vaccine Risk and Mistrust in Medical Institutions

Over half of the participants suggested that patient/self-perception of vaccine risk served as a contributor to COVID-19 vaccine hesitancy. However, the source of this perception ranged from misinformation to personal and community experiences.
“*[Patients] heard that somebody they know [had a] really bad reaction to the second dose, and now they’re extra nervous about it*,” Medical professional.
“*[People are] afraid that if they do take [the vaccine] they’re going to get sick*,” Caregiver.
“*There are a lot of scary stories out there about potential negative effects of vaccines*,” Caregiver and medical professional.

Medical professionals often indicated that personal history associated with phobia/bad experiences played a role in current vaccine hesitancy or willingness to visit a medical office for an intervention, vaccines included.
“*There’s just a subset of people out there that have these phobias, they might have had a bad experience*,” Medical professional.

### 3.3. Mental Health Challenges and Sensory Sensitivities

Some participants indicated the importance of physical or mental sensitivities to medical procedures as a decision-making factor related to vaccines. Of these, many more specifically described a phobia of needles as a major source of delayed vaccination or vaccine hesitancy.
“*So, for kids, for example, who might be afraid of vaccines or maybe have needle phobia*,” Medical professional.
“*Individuals with neurodevelopmental disabilities may have sensory concerns, and so because the vaccination involves a needle that does puncture the skin. And on top of the anxiety of neurosensory issues this population may have increased challenges with receiving a vaccination or getting blood drawn*,” Medical professional.

Participants additionally discussed the contribution of mental health as a roadblock to vaccination.
“*You know, health care, anxieties, and even coming to the clinic, may be a little bit anxiety provoking for them… I think during a pandemic, where they can perceive it’s like ‘Oh, what if I catch an infection, if I you know, go to the clinic?’*” Medical professional.

However, anxiety or depression symptoms that contribute to vaccine hesitancy were often discussed within the context of a specific factor that triggered those symptoms, rather than hesitancy due to a generalized anxiety or depressive disorder [27].

### 3.4. Other Barriers: Structural Barriers and Lack of Awareness

Especially for individuals in the ND community, accessibility concerns can serve as a barrier to obtaining necessary medical procedures in a timely manner. Specifically, for COVID-19 vaccines, incorporating accessible measures that enable ND individuals to obtain all necessary procedures and information can mean the difference between early and safe vaccination.
“*Some of it is like the even like this room that I’m in right it’s not built for anybody with a wheelchair*,” Medical professional.

Healthcare professionals additionally expressed concerns over access to transportation and time that would allow individuals to obtain their COVID-19 vaccines according to the approved schedules.
“*It’s having the Kroger pharmacy, or having the CVS and the Walgreens, which right now all have the signs up saying walk in, you know, COVID vaccinations are available, the problem is you got to be able to get to that location to walk in, and they’re all more than two miles away from most people’s home*,” Medical professional.
“*But I think we’ve seen some more barriers in terms of actually getting them into the clinic to do the vaccines*,” Medical professional.
“*I had a patient the other day that said no, I’ve been sick for the last three months and I haven’t been able to drive, and I don’t have a ride*,” Medical professional and caregiver.

### 3.5. Facilitators to Vaccine Confidence

Most participants cited that reference to reputable sources of information is a beneficial factor in discussing vaccine hesitancy (Table 3). For instance, for medical professionals, pointing patients towards legitimate and scientifically accurate resources on COVID-19 and vaccine information enabled vaccine confidence.
“*I always mentioned that we like to follow the CDC vaccine guidelines and recommendations*,” Medical professional.
“*I refer them to go back to the CDC*,” Non-medical health professional.
“*I’ve been providing evidence-based resources*,” Medical professional.

On the patient end, interviewees suggested that access to accurate and up-to-date information influenced how they felt about COVID-19 vaccines.
“*[The] CDC started helping a little bit and made me more confident*,” Caregiver.

The subsequent dissemination of more complicated vaccine data was recognized as another strategy that providers can implement with vaccine hesitant patients. Often, individuals with lived experience as patients or loved ones of patients requested more dissemination of scientific information from medical professionals.
“*I think that actually showing like real data, or a subset of data, that sort of boil down is good*,” ND-identifying physician in training.
“*Give as much information and as clear of a way that you could, so that parents would be more likely to get it*,” ND caregiver.

When working with young ND patients, breaking down COVID-19 information in an age-appropriate manner not only improves family comfort with vaccines, but can also ease barriers to adherence with COVID-19 protocols. For instance, one caregiver suggested that proper dissemination in conjunction with individual approaches would have benefitted their loved ones alongside many others.
“*A lot of people with special needs, they go by pictures, not just words, they want pictures, they want simple language*,” ND caregiver.

Often, these discussions can be highly effective when conducted by trusted primary care providers, as they can serve as the main source of healthcare patients receive.
“*I recommend their primary care because at the end of the day that’s their central person*,” Non-medical health professional.
“*I think one of the best things now…. is that things have kind of trickled down to individual providers with their specific patients*,” Medical professional.

### 3.6. Suggestions for Improving Vaccine Confidence

Regarding the approach by which to target vaccine hesitancy in patients, physicians and healthcare professionals often questioned the approaches used to communicate with patients who are vaccine-hesitant. However, they still acknowledged the important role that close, positive communication played in explaining the cost–benefit analysis of COVID-19 vaccination.
“*Rich, full conversation and that openness is really important, and many people, I think, who initially [are hesitant]*,” Non-medical Health Professional.
“*You know, [it] makes a big difference to being able to … Say, ‘Hey, Tell me your concerns. Tell me what your questions are.’ Let’s try to see what we can do to answer those questions*,” Medical professional.

As a complement to improved communication strategies, the amount of time a medical professional spends with their patient to ease their concerns can play a major role in a patient’s perception of medical interventions. Especially for ND patients, incorporating extra time in an appointment to discuss a more individualized approach to vaccines can improve confidence in said vaccines.
“*Scheduling those patients for a little bit of extra time, maybe doing that prep work ahead of time*,” Medical professional.
“*A typically shot [is] like 10 min or 15 min. You might consider extending that time for someone who is more anxious*,” Medical professional.

Such programs would include medical or non-medical accommodations. A non-medical accommodation includes distraction tools such as the “buzzy bee,” which distracts the patient from the pain of a needle puncture by vibrating against the patient’s skin. Other suggestions included scheduling patients with needle anxiety for a longer appointment time to allow care providers to thoroughly explain the procedure and create a calm and relaxed environment. Another interviewee suggested offering to sedate or provide local anesthetic, such that the patient does not feel any pain during the vaccination. Such accommodations can mitigate patient anxieties and fears about vaccination. 

However, strategies to approach vaccine hesitancy often go beyond the confines of the doctor–patient relationship. Many participants highlighted the importance of tackling vaccine hesitancy within individual family units and communities.
“*Also just…advocating for either yourself as the patient or for the patient if you’re a parent*,” Medical professional.
“*I think we really need to get into the community and do more work internally from inside out, as opposed to top down*,” Social worker.

## 4. Discussion

Previous research has shed light on the barriers that individuals face in acquiring COVID-19 vaccines alongside the main factors contributing to vaccine hesitancy. This study uniquely contributes to the evidence base by further investigating hesitancy within the context of identifying strategies to increase COVID-19 vaccination rates as they pertain to ND communities. The detailed responses from stakeholders in the ND community to questions related to vaccine hesitancy reflect a major strength of our work and qualitative approach. The following conclusions are derived from stakeholder suggestions to increase vaccine confidence in ND communities.

Many of the major barriers we report, such as mistrust or misinformation, have been previously implicated in vaccine hesitancy literature [3,10,25,28]. Interviewees cited that the misinformation about the efficacy and side effects of the COVID-19 vaccine at the beginning of the pandemic was concerning. Our work reinforces the role that misinformation can play in vaccine hesitancy within ND communities. Correcting misinformation and recommending reliable sources of information to patients and their caregivers is a major focus for interviewees working in healthcare services. 

Another major contribution of our work is in the significance of sensory sensitivities in contributing to vaccine hesitancy. Particularly within the ND community, sensory sensitivities, such as needle anxiety or loud, fast paced environments, have contributed to vaccine hesitancy [29]. As such, it is crucial to address sensory sensitivities in vaccine confidence programs to ensure that vaccines are distributed in a timely manner to ND communities.

A major benefit of our work is in our capacity to obtain an in-depth view of the appropriate steps needed to tackle vaccine hesitancy. Particularly for the ND community, offering appropriate accommodations at vaccination sites can make a major difference in vaccine adherence within this community. Additionally, ensuring equal access to information regarding the vaccination process, and offering forthright and clear instructions on what ND patients should expect when getting vaccinated is a major contributor to improved trust and comfortable vaccination experience. Our work suggests that guiding patients towards accurate and recent data on COVID-19 vaccine safety and efficacy can contribute to improved confidence in these vaccines.

In addition to necessary accommodations, structural change, and public health programs, a major factor that can impact patients who are vaccine hesitant is improved communication with physicians and medical professionals. Previous studies have also suggested that receiving information from healthcare workers help patients to develop vaccine awareness, thus minimizing misinformation, and that concerns are magnified due to a lack of clear communication [20,30]. When discussing vaccines with hesitant patients, physicians engaging in an open dialogue in which patients can express their concerns without judgment tend to be more effective in damping those concerns and discussing vaccines in a more positive manner.

While the interviews were focused on COVID-19 vaccine hesitancy, many of the applications of our results can be applied to broader vaccine hesitancy or preventative care. For instance, the removal of structural barriers to accessing healthcare and vaccine centers can improve uptake in adherence to yearly influenza vaccination protocols. Initiatives such as the Needle Anxiety Program at UCLA, while proven to be useful for vaccination, were developed to enable needle-anxious patients to receive necessary blood tests in a timely manner.

There are multiple limitations to consider regarding our work. Given the qualitative nature of our exploration into vaccine hesitancy, our analysis does not comprehensively cover the underlying reasons for COVID-19 vaccine hesitancy within the United States or abroad. Our study is considered a pilot study that collected information on a small scale, with the intention that findings could be utilized on a larger scale with additional research. Our sample population primarily had connections to the UCLA health system, and, as such, are neither nationally nor internationally representative of ND communities. Physicians were overrepresented as compared to ND individuals. Instead, we focused on identifying individuals who strongly identified with the ND community, or were often involved in multiple capacities, such as through individual or family experience and profession. Lastly, it should be noted that face-to-face interviews may yield a tendency to satisfy social desirability amongst participants [31].

## 5. Conclusions

This research highlights how public health programs aimed at increasing vaccine confidence can be improved to address the concerns of ND populations. Our findings revealed that the vaccine hesitancy of ND populations may be attributed to a lack of accommodation. The results show that future work on increasing vaccine confidence within the ND community should include research on effective dissemination of public health programs to vulnerable communities, and the implementation of accommodations for ND patients in healthcare settings.

## Figures and Tables

**Table 1 vaccines-11-00895-t001:** Interviewee roles.

Field	Position	Number
Healthcare	Physician	7
	Nurse	1
	Social Worker	1
	Physician in Training	2
Academia	Science Faculty (Non-Medical Health Professional)	6
Lived Experience	Caregiver *	3
	ND Individual	2

Caption 1. In this table, we describe the demographics of interviewees by their position or relationship to the ND community. Positions were not mutually exclusive. Five interviewees held two positions simultaneously. * Caregiver is defined as an individual who provides a range of support for a ND-identifying individual. All caregiver interviewees were family members of ND individuals.

**Table 2 vaccines-11-00895-t002:** Qualitative results of barriers to confidence.

Code	Operational Definition	Example Quotations	% Frequency	Occurrence
Lack of awareness	Lack of awareness of programs that support vaccination efforts	“I think there’s a lot of people who aren’t aware of the service that we offer at the needle anxiety program.” —Medical professional	37.50%	6
Misinformation	General misinformation or misconception perceived by people	“It can be really scary sometimes because of the conviction of which some people believe fantastical things.” —Non-medical health professional	81.25%	13
Mistrust in medical system/personal/community experiences	Mistrust or phobia that prevents vaccination either due to historical discrimination or past experiences with the medical system	“In medical settings, if you’ve been in that situation where you’ve never been heard, or where someone told you ‘We’re going to one thing,’ but really, they do something else to you quickly, you know that’s what will stay with you.” —Medical professional	43.75%	7
Perception of vaccination risk	Any perception that vaccines pose a risk to their health. Could be due to misinformation or misconception	“There is a fear of getting an illness that you can prevent with a vaccine or fear of getting a vaccine that may have unknown side effects.” —Medical professional and caregiver	56.25%	9
Mental health/sensory sensitivities or other activities	Mental health or sensory sensitivities that prevent people from receiving vaccines or adhering to COVID-19 protocols	“Individuals with neurodevelopmental disabilities may have sensory concerns, especially with vaccination involving a needle that punctures the skin leading to increased challenges with receiving vaccination.” —Medical professional	87.50%	14
Structural/socioeconomic barriers	Structural barriers that prevent people from receiving vaccines	“Those are also the same neighborhoods where people don’t have independent vehicles… they’re quite dependent upon bus lines and the bus lines are mediocre at best.” —Social worker	37.50%	6

Caption 2. Qualitative results for barriers to promote vaccine confidence by category, operational definitions, example quotation, and percent frequency, and occurrence defined as the number of interviews the theme occurred in.

**Table 3 vaccines-11-00895-t003:** Qualitative results of facilitators to vaccine confidence.

Code	Operational Definition	Example Quotation	% Frequency	Occurrence
Community-based strategies	Community-targeted strategies that facilitate COVID-19 vaccination or vaccine confidence	“Our regional center [sent] some emails and phone calls along the way [that the patient is too young and not yet eligible for a vaccine for.” —Caregiver and medical professional	37.50%	6
Knowledge and information dissemination	Information that is disseminated regarding COVID-19 broadly or the vaccine that helps people to understand and receive vaccination	“Information [was] disseminated about [how] even though [the vaccine] came out very quickly, it was decades in the making, and that this technology has been [thoroughly] worked on and developed.” —Medical professional	50.00%	8
Targeted social media campaigns	Modes for targeted social media campaigns that have been successful in facilitating vaccine confidence	“[We use] different voices or different styles [depending on] the nature of the audience and it’s been really helpful for us to make new connections.” —Non-medical health professional	18.75%	3
Perceived importance of vaccination	Perception that vaccination is good for themselves, family, and society	“It was very reassuring that at some point she will also be protected from something that could be significantly more dangerous for her than her siblings.” —Caregiver and medical professional	37.50%	6
Providing medical accommodations	Medical therapies or treatment used as accommodations for the general public or neurodivergent community	“We can administer between 5 to 10 milligrams of sedation [for needle anxiety].” —Medical professional	31.25%	5
Providing non-medical accommodations	Non-medical accommodations provided to the general public or neurodivergent community	“Whatever is going to help calm [the patient can help with the vaccination process]. Some distractions might be something a buzzer or videos.” —Medical professional	31.25%	5
Current public health/vaccine policies and programs	Broad public health vaccination programs or policies that make it easier for people to get vaccination	“Decreasing the anxiety around a procedure that involves needles would help everyone with this concern and not just disabled populations.” —Medical professional	25.00%	4
Experience or relationship with a health care professional	Interpersonal relationship or experience with healthcare professional that facilitates vaccine confidence	“Doctors talk to their patients about needle anxiety program, so their [positive] experience with the doctor allows them to get the vaccine in a way that’s comfortable.” —Medical professional	37.50%	6
Sources of information	Broader sources of credible public health information and news used by experts	“Folks must be cautious with where they are getting their information and ensure that they are validated sites that are scientific.” —Non-medical health professional	56.25%	9
Subjective norm	Seeking normalcy or following perception of what others believe is right	“I sense a large decision maker for a lot of people is wanting to get to a life of some degree of normalcy.” —Caregiver and medical professional	25.00%	4

Caption 3. Qualitative results for facilitators to promote vaccine confidence by category, operational definitions, example quotation, and percent frequency.

**Table 4 vaccines-11-00895-t004:** Qualitative results for suggestions to promote vaccine confidence.

Code	Operational Definition	Example Quotation	% Frequency	Occurrence
Bridging gaps in knowledge of vaccine	Ways to improve transparency and information dissemination between scientific community and general public	“Initially we focused on making sure people understood what a vaccine is and that it’s not like a magic bubble.” —Non-medical health professional	18.75%	3
Public health/vaccine policies and programs suggestions	Public health vaccination programs or policies that make it easier for people to receive vaccination and that are widely applied	“A targeted approach and maybe increasing resources in some of those areas where the families just aren’t really going to be able to access things or advocate for things on their own.” —Social worker	18.75%	3
Suggestions for future research	Future research ideas or suggestions	“I think that we should have additional studies to that should be conducted on vaccines for different populations.”	12.50%	2
Suggestions for families and caregivers	Ways families can help themselves or their community become more confident in the COVID-19 vaccine	“Family members who bring in the patient are also a source of support and calm and understands exactly what’s going to happen and is well coordinated, [this] reduces the patient’s anxiety.” —Medical Professional	50.00%	8
Communication	Suggestions for improving patient–provider communication	“I’d say the most important thing is nobody likes surprises. The more we can explain up front, maybe you walk through the scenario, talk through how it’s going to go, the smoother things will go.” —Medical Professional	75.00%	12
Medical/epidemiology knowledge	Suggestions for health care providers to improve their own medical knowledge (re: vaccinations)	“So, what [colleague physician] did is she went and downloaded all the articles. Now, she wasn’t saying to the patients that they need to go read all those articles. She’s basically saying, I know how to read all those articles. And I read them all and I can tell you about it.” —Non-medical health professional	12.50%	2
Training	Recommendation for improving the training of health care professionals and students in training	“We have implemented some training to have our medical assistants take a smidgeon longer with some of these anxious patients because maybe that’s all they need.” —Medical professional	31.25%	5
Accommodations designed to remove of structural barriers to vaccination access	Accommodations to alleviate structural or socioeconomic barriers that prevent people from receiving vaccines	“So, you’re being thoughtful that you know, this needs to be addressed and maybe revisit periodically to check in is important, maybe scheduling those patients are a little bit of extra time, maybe doing that prep work ahead of time during email or something like that.” —Medical professional	50.00%	8

Caption 4. Qualitative results for suggestions to promote vaccine confidence by category, operational definitions, example quotation, and percentage frequency.

## Data Availability

Interviews are publicly available; however, the qualitative data presented in this study are available on request from the corresponding author.

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
