# Peer review of "A Qualitative Investigation on COVID-19 Vaccine Hesitancy in Neurodivergent Communities"

_vaccines, 2023, doi:10.3390/vaccines11050895_

Round 1
Reviewer 1 Report
This manuscript reports the results from a qualitative study on barriers, facilitators and recommendations towards COVID-19 vaccine hesitancy in a particularly vulnerable population represented by Neurodivergent individuals. The methodology used is in-depth interviews.
The general reporting of the manuscript is good and easy to read. The introduction in well organized and the conclusions appropriately drawn.
I have just some minor comments.
1. In the interviewed sample the physicians in particular, but also professionals in general, are over-represented than the ND individuals. This should be reported in the limitations.
2. Table 2,3 and 4. Since the number of interviewed is low (n=17), it could be useful to report in the last column also the absolute number (in addition to the percentage)
3. As the Authors recognized one study limitations are the qualitative nature of the study and a sample of individuals not representative of ND communities than this study is explorative. Given these observations I think that it is better to talk about "suggestions" instead of "recommendations". Recommendations could lead one to think about guidelines that must be based on specific and standardized methodology.
Author Response
We greatly appreciate your review of our manuscript and have enjoyed working with the Vaccines team throughout our reviews. Please see the attached document for point-by-point responses.

Reviewer 2 Report
Dear editor,
Thank you for the kind invitation to review this manuscript. Attached are my comments for the authors' consideration.
The introduction is generally well written. I suggest to add a comment on the importance of exploring the point of view of health care operators as they are the first point of contact for patients and families and one of the most trusted source of information on vaccine. Thus they represent a privileged point of view
Suggested reference:
Alderotti, G.; Corvo, M.F.; Buscemi, P.; Stacchini, L.; Giorgetti, D.; Lorini, C.; Bonaccorsi, G.; Pinilla, M.J.C.; Lastrucci, V. Communicating with Patients about COVID-19 Vaccination: A Qualitative Study on Vaccinators in Tuscany Region, Italy. Vaccines 2023, 11, 223.
Schmitt, H.-J.; Booy, R.; Aston, R.; Van Damme, P.; Schumacher, R.F.; Campins, M.; Rodrigo, C.; Heikkinen, T.; Weil-Olivier, C.; Finn, A.; et al. How to optimise the coverage rate of infant and adult immunisations in Europe. BMC Med. 2007, 5, 11.
Methods
- In line 106 and 107, it says that interviews were conducted between July and September 2021 and between July and September 2022, then you state that the project was approved in March 2022 (line 378). Please review and correct as appropriate.
- Please report your methods in line with the COREQ checklist for reporting qualitative study
- Although most of the necessary information are reported in the methods section, they are presented in a confused and unordered way. Please revise the order of methods presentation changing paragraphs order and name, here an example of possible methods’ paragraphs: Study setting and Design; Population; Data Collection, Processing and Analysis.
- Please better specify the recruitment process applied and if there were any inclusion and exclusion criteria for the study population. Please also specify how informed consent was obtained by ND participants
- Please provide demographics of the participants, a supplementary table could be used.
- Some information is redundant (e.g. line 101-102 and 116 -117).
-Please specify to which participant categories were used the interview guides. Furthermore, it is not clear if one of the provided interview guide were used also for ND participants. In case the interview guide is not provided, please provide it. Lastly, in Appendix A1 almost all the questions are referring to a “Needle Anxiety Program at UCLA” please provide more information on this program and on why it is relevant for the study aims in the methods or in the introduction sections.
Results
- Is there any difference between caretaker and caregiver (line 263-256-248-236-177)? If not, please choose one of the two terms and use it consistently throughout the manuscript.
Discussion
-It would be helpful to better discuss your results compared to findings from previous studies. For example, it would be interesting a comparison with other strategies identified by first-line healthcare workers to increase vaccination confidence during the Covid-19 pandemic reported by previous studies.
I suggest you some studies.
Alderotti, G.; Corvo, M.F.; Buscemi, P.; Stacchini, L.; Giorgetti, D.; Lorini, C.; Bonaccorsi, G.; Pinilla, M.J.C.; Lastrucci, V. Communicating with Patients about COVID-19 Vaccination: A Qualitative Study on Vaccinators in Tuscany Region, Italy. Vaccines 2023, 11, 223. https://doi.org/10.3390/vaccines11020223
Reiter, P. L.; Pennell, M. L.; Katz, M. L. Acceptability of a COVID-19 Vaccine among Adults in the United States: How Many People Would Get Vaccinated? Vaccine 2020, 38(42), 6500–6507. https://doi.org/10.1016/j.vaccine.2020.08.043.
- As the study is based on data from face-to-face interviews, a social desirability tendency of participants cannot be dismissed, so it is necessary to explore this aspect as a limit of your study
- The sample size is small even for a qualitative study, especially considering the fact that different categories of participants were considered. However, the results are interesting as the topic explored is novel. Please highlight that the study is a pilot and underline the limited sample size in the limitation section
Author Response
We thank you for your review of our manuscript and for your service in improving our work. Please see the attached document for our point-by-point responses.

Reviewer 3 Report
Dear editor,
Thank you for the kind invitation to review this manuscript.
Below are my comments
Abstract
- Will be helpful to define what neurodivergent refers to.
Introduction
- It will be helpful to highlight high rates of vaccine hesitancy among the general population and the lack of studies in neurodivergent patient population
-> To cite the following article: https://pubmed.ncbi.nlm.nih.gov/34452026/
Methods
- Suggest to adhere COREQ checklist for reporting of the study
- Given the general broad spectrum of neurodivergent population and the recruitment of only 17 individuals, did the authors manage to cover the entire spectrum of neurodivergent patient population related comorbidities.
- Given the title focused on neurodivergent populations, I was puzzled whay the authors included individuals from healthcare settings.
-> Ideally to answer the hypothesis, instead of assessing individuals who have only have indirect relationship with ND populations, it would be better to include only participants from ND communities
Results
- generally well written
Discussion
- What are the unique findings in the study?
- How does it compare with existing studies in the general population and are there specific factors identified with regards to ND population
- What are the implications of this study and should policies be written for this population?
Author Response
We would like to thank Reviewer 3 for their insightful comments and revisions of our manuscript. We appreciate the opportunity to improve our work and look forward to further communications with the Vaccines team. Please see the attached document for point-by-point responses.

Round 2
Reviewer 2 Report
The authors have accommodated the suggestions
Reviewer 3 Report
nil further comments